# Probabilistic Salient Object Ranking

Rongjin Guo [1]   Huankang Guan[✉ 1]   Rynson W.H. Lau[✉ 1 2]

## Abstract

Salient Object Ranking (SOR) aims to study how humans visually explore complex scenes by predicting an ordered sequence of objects that attracts our attention. Existing SOR approaches typically model this ranking deterministically, assuming a single, fixed ranking sequence of attention. However, such deterministic SOR fails to capture the true nature of human attention. We observe that human attention shifts exhibit variability and stochasticity, *i.e.*, the next object of fixation is not a definitive choice but rather a probability distribution. Yet, existing SOR methods and evaluation metrics do not account for this inherent randomness. To address this fundamental problem, we first propose ProbSOR, a novel Probabilistic Salient Object Ranking framework built upon a vision–language model (VLM) backbone. By incorporating Group Relative Policy Optimization (GRPO), ProbSOR explicitly learns the uncertainty of attention shifts. We then propose a new metric tailored for ProbSOR, as existing SOR metrics only support deterministic rankings. We further construct a ProbSOR dataset comprising 15,000 probabilistic SOR samples, to support both model training and evaluation. Extensive experiments show that ProbSOR achieves strong performance in salient object ranking under both our proposed and traditional benchmarks. The code is available here.

## 1. Introduction

Understanding how humans shift their attention when exploring complex scenes has been a long-standing research problem (Johnston & Dark, 1986; Koch & Ullman, 1987).

Salient Object Ranking (SOR) is recently proposed to study this process by identifying not only *which* objects are salient, but also *in what order* they attract humans' attention. This modeling of attention shift provides a more fine-grained understanding of human visual behavior, benefitting applications like gaze prediction (Nakashima et al., 2015; Cerf et al., 2007), autonomous driving (Sachdeva et al., 2024; Fu et al., 2024; Huang & Wang, 2024), and scene understanding (Li et al., 2023; Du et al., 2019; Zhang et al., 2014).

In recent years, we have witnessed a remarkable progress in SOR research. Islam et al. (Amirul Islam et al., 2018) first extended traditional salient object detection (SOD) to simultaneously detect, rank, and subitize multiple salient objects within a scene. Later, Siris et al. (Siris et al., 2020) redefined SOR from a psychological perspective, interpreting saliency as the sequential shift of human attention across objects. Subsequently, some methods (Sun et al., 2023; Liu et al., 2021a) focus on structural modeling by introducing various architectural innovations to capture inter-object dependencies and contextual hierarchies. Another method (Guan & Lau, 2024b) incorporates psychological priors, improving the accuracy of ranking predictions by mimicking the human attention mechanism. Several studies (Wu et al., 2024; Zhang et al., 2025) emphasize saliency-related cues, integrating features like shape, texture, distance to enhance the salient object representation. Other efforts (Guan & Lau, 2024a; Liu et al., 2025a) explore fusing multi-modal signals like linguistic or pose to move SOR beyond the purely visual domain. Despite the contributions from different perspectives, all these methods share a common limitation: they learn to predict a *deterministic* ranking and evaluate against an exact ground-truth order using metrics such as SOR Score (Siris et al., 2020) and SA-SOR (Liu et al., 2021a). Such deterministic modeling inherently assumes *one single correct ranking*, and may penalize other equally plausible fixation sequences as errors even though they may still represent valid attention patterns.

In this work, we observe that human visual attention is inherently probabilistic. The next object that attracts our attention may not always be the same, even for the same person, but is probabilistically influenced by the prior viewing trajectory, scene context, and perceptual biases, as supported by extensive findings in cognitive psychology (Peelen & Kastner, 2014; Engbert et al., 2015; Johnston & Dark, 1986;

[1]City University of Hong Kong, Hong Kong SAR, China [2]City University of Hong Kong, Dongguan, China. Correspondence to: Huankang Guan <huankguan2-c@my.cityu.edu.hk>, Rynson W.H. Lau <RynsonLau@gmail.com>.

*Proceedings of the 43rd International Conference on Machine Learning*, Seoul, South Korea. PMLR 306, 2026. Copyright 2026 by the author(s).

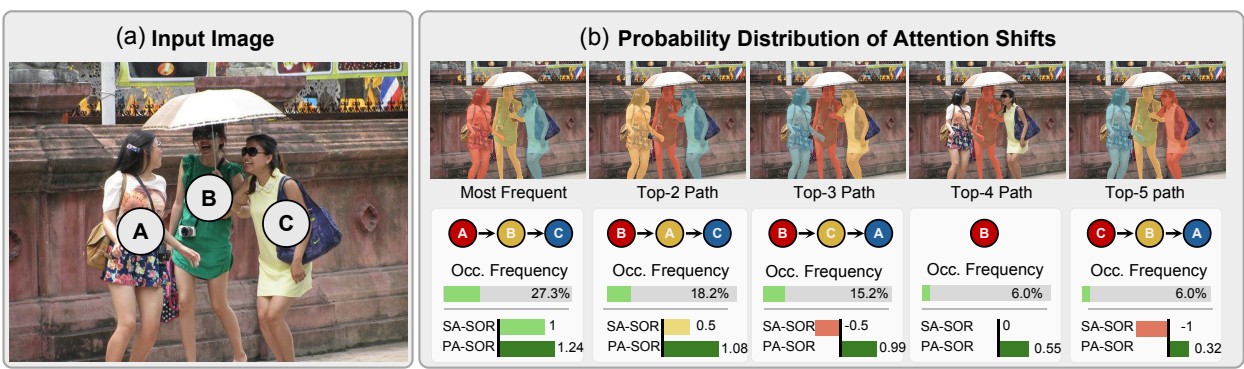

*Figure 1.* **Illustration of diverse human attention-shift patterns:** (a) input image, and (b) top-5 most popular attention-shift patterns obtained from the SALICON dataset and their corresponding occurrence frequencies. Assuming the most popular pattern (with the highest occurrence frequency) as the ground truth under the existing evaluation metric (SA-SOR (Liu et al., 2021a)) may unfairly penalize other less frequent but plausible sequences. In contrast, our probabilistic metric, PA-SOR, rewards each sequence by its popularity, enabling fair and consistent assessment across diverse attention behaviors.

Koch & Ullman, 1987). To further substantiate this observation, we re-examined the SALICON dataset (Jiang et al., 2015), which provides human fixation trajectories on natural images. As illustrated in Figure 1(b), although different viewers may have different ranking orders, the most popular order is typically taken as the single ground truth in existing SOR formulations. As a result, any prediction that deviates from this ground truth sequence, even if it is still a plausible one, would be treated as incorrect and assigned a low or even negative score. Overemphasizing on one "correct" shift order unfairly penalizes plausible variants, leading to overfitting to specific annotation orders while ignoring the intrinsic stochasticity of human attention.

To address this limitation, in this paper, we propose **Prob**abilistic **S**alient **O**bject **R**anking (ProbSOR), which models SOR as a sequence of conditional probability transitions among salient objects. Specifically, this attention-shift sequence can be represented as a probability tree, which we refer to as the *Attention-Shift Probability Tree* (ASPT). Nodes in an ASPT represent salient objects, while edges encode the conditional transition probabilities. The root node connects to all plausible starting objects. Each path from the root node extends until reaching a termination node that marks the end of the sequence. It corresponds to a valid attention-shift sequence with an associated probability and reward. This formulation enables explicit modeling of the uncertainty (distribution) in human attention shifts. To learn this distribution over attention-shift paths, we leverage a Vision-Language Model (VLM) as the backbone and optimize it with a two-stage training pipeline. First, we establish the model's fundamental ranking capability via Supervised Fine-Tuning (SFT) on the most probable attention paths. To capture the intrinsic stochasticity of human attention, we then draw inspiration from the Group Relative Policy Optimization (GRPO) algorithm used in fine-tuning Large Language Models (LLMs), and adapt its group-based update

strategy to align the model's predictions with the probability distribution over diverse attention-shift paths, thereby encouraging the model to produce diverse yet behaviorally plausible human-like patterns.

We further design a new metric, Probability-Aware SOR (PA-SOR), for evaluation purposes. Instead of comparing predictions against a single ground-truth ranking, our metric measures the plausibility of a sequence by accumulating the rewards based on transition probabilities along the path, yielding a more reliable and fair assessment across diverse attention-shift patterns. Finally, to facilitate model training and evaluation, we have constructed a ProbSOR dataset by restructuring SALICON data into probabilistic attention trajectories. This dataset provides 15,000 samples with multiple ranked attention-shift sequences per image. Extensive experiments demonstrate that our ProbSOR achieves strong and consistent improvements over deterministic SOR methods, validating its ability to model the variability and uncertainty of human visual attention.

In summary, our key contributions are as follows:

- We reformulate Salient Object Ranking into a probabilistic framework (ProbSOR) that captures the stochasticity and diversity of human attention shifts.

- We propose a learning framework, which is based on Group Relative Policy Optimization (GRPO), to explicitly model the uncertainty in attention transitions, and a new probabilistic evaluation metric for assessing diverse attention shift sequences.

- We propose a new dataset with probabilistic attention trajectories for model training and evaluation.

- Extensive evaluations show the effectiveness of the proposed method on our proposed dataset, demonstrating its ability to model the stochastic nature of human attention,

and on existing SOR datasets, demonstrating its generalization ability.

## 2. Related Work

### 2.1. Salient Object Ranking (SOR)

SOR aims to model human attention shift across salient objects in a scene. Islam et al. (Amirul Islam et al., 2018) first introduced the task by extending saliency detection to jointly detect, subitize, and rank salient objects. Siris et al. (Siris et al., 2020) then draw on psychological theories of attention to reinterpret SOR as a sequence of attention shifts across objects. Subsequent studies (Fang et al., 2021; Liu et al., 2021a) focus on object-level reasoning, modeling inter-object dependencies to predict saliency ranks. Recognizing the limitations of purely object-level interactions, further research evolves to incorporate richer contextual information. OCOR (Tian et al., 2022a) models region-level interactions via spatial attention. Graph-based methods capture contextual relations through hierarchical structures (Deng et al., 2024), scene-level graph modeling (Qiao et al., 2024), and shape–texture disentanglement (Wu et al., 2024). Guan and Lau (Guan & Lau, 2024a) leverage human pose as a cognitive cue, while SeqRank (Guan & Lau, 2024b) simulates sequential attention shifts by modeling foveal and peripheral vision. Most recently, LG-SOR (Liu et al., 2025a) leverages language-guided reasoning from large vision-language models, and SOR-RL (Gao et al., 2025) formulates the task as a reinforcement learning problem to mimic human attention policies.

Despite their success, all these methods formulate SOR as predicting a deterministic sequence of salient objects. This simplification ignores the variability and stochasticity of human attention.

### 2.2. Salient Object Detection (SOD)

SOD forms the basis of SOR. It aims to generate a pixel-level map of the most visually prominent regions in an image (Cheng et al., 2014; Klein & Frintrop, 2011; Perazzi et al., 2012; Achanta et al., 2009; Siris et al., 2021; Wang et al., 2023; Wei et al., 2020; Zhang et al., 2019; Tian et al., 2023; Li et al., 2024). Earlier SOD method heavily rely on hand-crafted features like contrast and edges (Cheng et al., 2014; Klein & Frintrop, 2011; Perazzi et al., 2012; Achanta et al., 2009), which are later superseded by deep-learning models (Liu et al., 2021b; Siris et al., 2021; Wei et al., 2020; Zhang et al., 2019; Veksler, 2023; Tian et al., 2023). These advanced models improve the performance by incorporating strategies such as multi-scale feature fusion (Liu et al., 2021b; Wang et al., 2023), attention mechanisms (Liu et al., 2018; Zhang et al., 2018), and context aggregation (Siris et al., 2021; Zhang et al., 2019) to boost semantic awareness.

However, since SOD does not differentiate individual objects belonging to the same class, a different branch of research on **Salient Instance Detection (SID)** is proposed (Fan et al., 2019; Wu et al., 2021; He et al., 2017; Tian et al., 2022b; 2020; 2024). SID focuses on separating the salient objects in a scene into salient instances. Early methods largely rely on two-stage architectures (He et al., 2017), utilizing region proposal networks to first detect objects followed by learning discriminative features to distinguish salient instances from non-salient ones. To reduce reliance on costly pixel-wise mask annotations, recent works explore weakly-supervised learning (Liu et al., 2025b; Tian et al., 2022b; 2020) and unsupervised learning (Tian et al., 2024; Guan et al., 2025) paradigms.

Despite their success, neither SOD nor SID attempts to model the human attention shift across the detected objects.

## 3. Method

This section details our proposed ProbSOR method. Section 3.1 introduces the core theoretical framework and the *Attention Shift Probability Tree* (ASPT) to model visual stochasticity. Section 3.2 presents the model architecture, and Section 3.3 outlines our two-stage training strategy.

### 3.1. Probabilistic SOR Formulation

Existing SOR formulations treat the task as a deterministic sequence prediction problem, aiming to find a sequence of objects that represents the only "correct" attention-shift order. However, human attention shifts are inherently stochastic. The transition of attention from one object to another follows a conditional probability distribution, indicating that observers may explore a scene through diverse yet behaviorally plausible pathways. This variability implies that an image should admit multiple plausible attention sequences. However, prior formulations fail to accommodate this diversity, but instead, enforce a single ground-truth sequence.

To address this limitation, we reformulate Salient Object Ranking (SOR) into a probabilistic framework. Given an input image $I$, let $O = \{o_1, o_2, \ldots, o_N\}$ denote the set of salient objects detected from it. An attention-shift sequence is defined as:

$$\mathcal{S} = \{s_1, s_2, \ldots, s_T\}, \quad s_t \in O, \tag{1}$$

where $T$ is the sequence length, and $s_t$ represents the object attended at time step $t$. Each plausible sequence $\mathcal{S}$ carries a probability $P(\mathcal{S})$ that reflects how likely an observer is to follow that particular order when viewing the image. Hence, the goal of the SOR task should be to generate such plausible sequences $\mathcal{S}$ according to their underlying probabilities. We can model $\mathcal{S}$ as a Markov chain, where the transition to the next object is conditioned on the current state and the

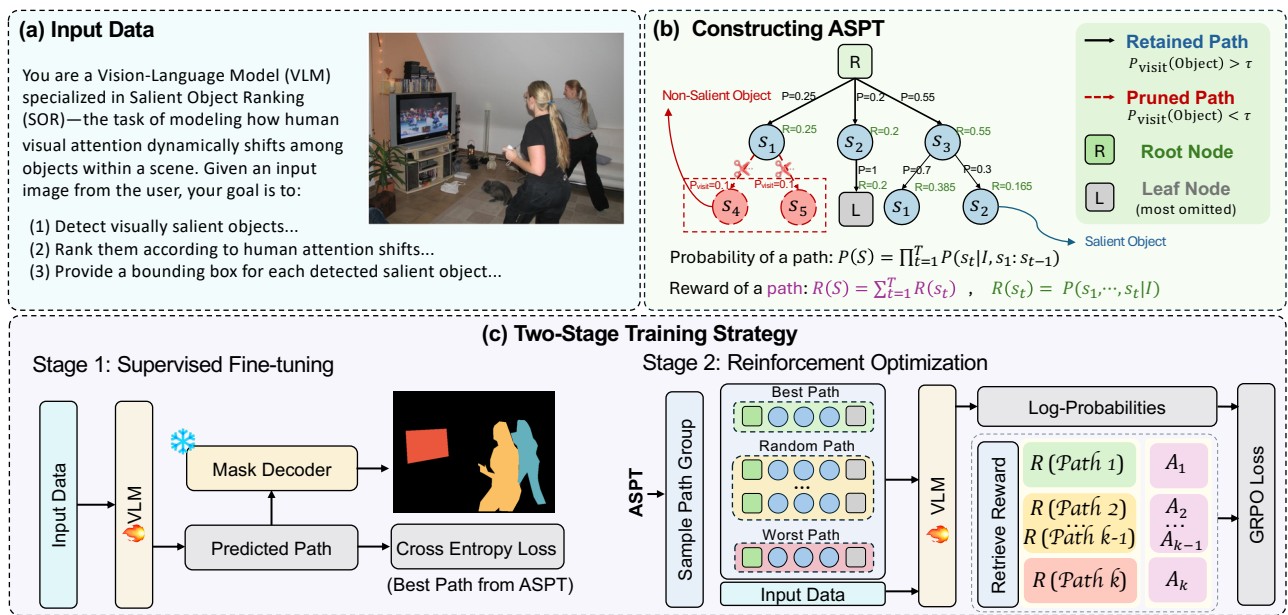

*Figure 2.* **Overview of the proposed ProbSOR framework. (a) Input Data.** The input image and prompt for the VLM. **(b) ASPT Construction.** Human fixation data are converted into an Attention-Shift Probability Tree (ASPT). Low-probability branches are pruned, and each sequence $\mathcal{S}$ is associated with its likelihood P(S) and reward R(S). **(c) Two-Stage Training Strategy.** Stage 1 performs supervised fine-tuning on the most popular attention shift sequence; Stage 2 samples multiple paths, $Path_1$, $Path_2$, ..., $Path_n$, and leverages ASPT-based rewards, $A_1$, $A_2$, ..., $A_n$, to optimize the model toward human-aligned attention-shift patterns.

previously attended objects. Therefore, the probability of sequence $\mathcal{S}$ given image $I$ can be defined as:

$$P(\mathcal{S} \mid I) = \prod_{t=1}^{T} P(s_t \mid I, s_{1:t-1}), \qquad (2)$$

where $P(s_t \mid I, s_{1:t-1})$ denotes the conditional probability of attending object $s_t$ given the image context and the attended history $\{s_1, s_2, ..., s_{t-1}\}$.

This probabilistic formulation can be equivalently represented as an **Attention Shift Probability Tree (ASPT)**. Each internal **node** in the tree corresponds to a salient object in the scene. The **root** node connects to all plausible first-fixation objects, whereas the final attended objects are linked to a terminal **leaf** node. Each **edge** encodes a conditional transition probability $P(s_t \mid I, s_{1:t-1})$, representing the likelihood of transitioning from a parent node to its child. Every complete **path** from the root node to the leaf node corresponds to a plausible attention-shift sequence $\mathcal{S}$, and the set of all possible sequences $\{\mathcal{S}\}$ constitutes the ASPT. Therefore, we model the human attention shifts as stochastic samples from the distribution $P(\mathcal{S} \mid I)$, with each sample corresponding to a unique path in the ASPT.

To assess the plausibility of an attention-shift sequence $\mathcal{S}$, we assign a **reward** to each $s_t$ based on how likely human observers are to attend to $s_t$ following path $\{s_1, \ldots, s_t\}$ given the image $I$. Specifically, we define the reward at step

$t$ as the probability of the visited prefix $s_{1:t}$:

$$R(s_t) = P(s_1, \ldots, s_t \mid I), \qquad (3)$$

This definition is intuitive since transiting to a node with a higher probability is more consistent with human attention behavior. This node should thus be given a higher reward. We define the reward of a sequence $\mathcal{S}$ as the sum of the rewards at all steps in $\mathcal{S}$, as:

$$R(\mathcal{S}) = \sum_{t=1}^{T} R(s_t). \qquad (4)$$

A higher $R(\mathcal{S})$ indicates an attention trajectory that better aligns with human attention behavior. Later, we use this reward formulation to guide the training in Group Relative Policy Optimization (Section 3.3) and to design our evaluation metrics (Section 5.1).

### 3.2. Model Architecture

Existing Vision-Language Models (VLMs) have demonstrated strong capabilities in visual reasoning, scene understanding and structured prediction. This makes them well-suited to serve as foundation models for various visual tasks. In our framework, we adopt Qwen3-VL (Bai et al., 2025; Yang et al., 2025) as the backbone, due to its strong multimodal reasoning ability and a favorable performance-efficiency trade-off. Figure 2(a) illustrates the inputs to the

model. Given an image $I$ and a prompt $\mathcal{Q}$, the model generates a structured textual output $\mathcal{Y}$, encoding an ordered sequence of salient objects:

$$\mathcal{Y} = \text{QwenVL}(I, \mathcal{Q}). \quad (5)$$

The output $\mathcal{Y}$ adheres to the following canonical format:

$$\mathcal{Y} = \left\{ \,|\, rank_t \,|\, category_t \,|\, bbox_t \,|\, \right\}_{t=1}^{T}, \quad (6)$$

where $rank_t$ denotes the rank order. $category_t$ is the object category. $bbox_t$ is the predicted bounding box $(x_{t,1}, y_{t,1}, x_{t,2}, y_{t,2})$. We then employ SAM 2 (Ravi et al., 2024) as the mask decoder. SAM takes as input the image $I$ and bounding box $bbox_t$, and generates a mask $\mathbf{M}_t$ for the $t^{th}$ salient object:

$$\mathbf{M}_t = \text{SAM}(bbox_t, I). \quad (7)$$

### 3.3. Two-Stage Training Strategy

To effectively train our ProbSOR model to capture the inherent stochasticity of human attention, we propose an efficient two-stage training strategy.

#### 3.3.1. STAGE 1: SUPERVISED FINE-TUNING (SFT)

We begin with the SFT to equip the VLM with task-specific generation capabilities. Specifically, we select the highest-probability path $\mathcal{S}_{\max}$ from the ASPT and transform it into a textual sequence that matches the format of $\mathcal{Y}$ (Eq. 6), using this sequence as the supervision signal for fine-tuning. We adopt the LoRA strategy to efficiently fine-tune the pre-trained Qwen3-VL backbone. This stage enables the model to accurately recognize and localize salient instances by generating structured output sequences.

#### 3.3.2. STAGE 2: GROUP RELATIVE POLICY OPTIMIZATION (GRPO)

We use the SFT model of Stage-1 as the initial policy $\pi_\theta$, and refine it via the GRPO to learn a reward-weighted distribution over multiple plausible attention-shift sequences. While Stage 1 teaches the model to reproduce the most likely attention-shift sequence, GRPO exposes it to multiple plausible sequences and optimizes the model to align with their relative rewards. In standard GRPO, groups of candidate sequences are obtained by sampling from the current policy. However, this leads to many low-probability or invalid attention paths in our task. As a result, each group will be largely filled with near-zero-reward sequences, making the GRPO optimization noisy and unstable. To mitigate this problem, we instead form groups by sampling candidate paths directly from the ASPT, which already encodes a well-structured distribution of human attention trajectories with informative rewards. At a high level, our goal is to learn a

policy $\pi_\theta$ that assigns higher probability to attention-shift sequences with larger ASPT rewards:

$$\max_\theta \; \mathbb{E}_{S \sim \pi_\theta(\cdot|I)}[R(S)]. \quad (8)$$

Specifically, we use a fixed group size $G$. For each image $I$, we keep the best- and worst-reward paths, and sample the remaining $G-2$ paths from the rest, yielding a compact group with balanced positive/negative signals for stable GRPO optimization. For each sampled sequence $\mathcal{S}_i$, we compute its log-probabilities $\log \pi_\theta(\mathcal{S}_i \mid I) = \sum_t \log \pi_\theta(a_t \mid s_t)$ and likelihood ratios $r_t(\theta) = \frac{\pi_\theta(a_t|s_t)}{\pi_{\theta_{\text{old}}}(a_t|s_t)}$. To obtain each sequence's relative merit within the group, we compute its advantage $\hat{A}_i$ by group-wise normalizing the rewards as:

$$\hat{A}_i = \frac{R(\mathcal{S}_i) - \mathbb{E}[\{R(\mathcal{S}_j)\}_{j=1}^G]}{\text{std}(R(\{\mathcal{S}_j\})_{j=1}^G)}. \quad (9)$$

We then optimize the policy to increase the likelihood of high-reward sequences while suppressing less favored ones:

$$\mathcal{L}_{\text{GRPO}}(\theta) = \frac{1}{G} \sum_{i=1}^{G} \frac{1}{|\mathcal{S}_i|} \sum_{t=1}^{|\mathcal{S}_i|} \Big\{ \min \big[ r_t(\theta)\,\hat{A}_i, $$
$$\text{clip}(r_t(\theta), 1-\epsilon, 1+\epsilon)\,\hat{A}_i \big] - \beta D_{\text{KL}}\big[\pi_\theta \,\|\, \pi_{\text{ref}}\big] \Big\}, \quad (10)$$

where $\epsilon$ and $\beta$ are hyper-parameters. The KL regularization term constrains the policy update to stay close to the initial distribution, preventing overly aggressive shifts in sequence generation. Through this optimization, the model gradually learns to approximate the underlying probability distribution over diverse attention-shift patterns, therefore capturing the stochasticity of human visual behavior.

## 4. Our Benchmark

### 4.1. Limitations of Deterministic SOR Evaluation

Existing SOR benchmarks, such as ASSR (Amirul Islam et al., 2018) and IRSR (Liu et al., 2021a), evaluate predictions against a single deterministic ranking, assuming that human attention follows one fixed sequence. This oversimplified assumption neglects the natural variability of human attention, penalizing other behaviorally plausible fixation orders and resulting in unfair evaluation. To address this limitation, we introduce **ProbSOR-Bench**, a new benchmark that represents human attention as a probabilistic distribution over multiple plausible object orders, providing a more faithful evaluation of attention-shift modeling.

### 4.2. ProbSOR Benchmark Construction

Our benchmark is built upon the large-scale **SALICON** dataset (Jiang et al., 2015), which contains dense fixation

*Table 1.* Quantitative comparison on ASSR and IRSR. Best results are marked in bold and second-best results are underlined. '–' indicates that the result is not available. We report two variants of our method. *Ours (One Stage)* denotes the model trained only with Stage-1 supervised fine-tuning on ASSR or IRSR training set. *Ours (Two Stage)* denotes the full ProbSOR model, which is first trained on ProbSOR-Bench with the two-stage SFT+GRPO pipeline and then directly evaluated on ASSR or IRSR without additional fine-tuning.

| Method | Reference | ASSR | | | | | IRSR | | | | |
|---|---|---|---|---|---|---|---|---|---|---|---|
| | | PA-SOR ↑ | PA-SOR$_{Norm}$ ↑ | IoU$_{mask}$ ↑ | MAE ↓ | SA-SOR ↑ | PA-SOR ↑ | PA-SOR$_{Norm}$ ↑ | IoU$_{mask}$ ↑ | MAE ↓ | SA-SOR ↑ |
| ASRNet (Siris et al., 2020) | CVPR'20 | 0.705 | 0.544 | 0.677 | 1.925 | 0.637 | 0.726 | 0.526 | 0.694 | 1.453 | 0.350 |
| PPA (Fang et al., 2021) | ICCV'21 | 0.742 | 0.568 | 0.678 | 2.019 | 0.647 | 0.736 | 0.536 | 0.716 | 1.924 | 0.501 |
| IRSR (Liu et al., 2021a) | TPAMI'21 | 0.755 | 0.575 | 0.640 | 2.205 | 0.643 | 0.782 | 0.569 | 0.680 | 2.257 | 0.545 |
| OCOR (Tian et al., 2022a) | CVPR'22 | 0.676 | 0.526 | 0.666 | 2.651 | 0.594 | 0.697 | 0.508 | 0.682 | 2.702 | 0.482 |
| PSR (Sun et al., 2023) | ACMMM'23 | 0.676 | 0.513 | 0.519 | 4.370 | 0.651 | 0.752 | 0.548 | 0.666 | 3.204 | 0.528 |
| SeqRank (Guan & Lau, 2024b) | AAAI'24 | 0.590 | 0.461 | 0.633 | 2.126 | 0.661 | 0.712 | 0.519 | 0.722 | 1.970 | 0.553 |
| QAGNet (Deng et al., 2024) | CVPR'24 | 0.844 | 0.652 | **0.796** | **1.203** | 0.772 | 0.820 | 0.601 | 0.777 | **0.923** | 0.618 |
| DSGNN (Wu et al., 2024) | CVPR'24 | 0.891 | 0.675 | 0.781 | 1.559 | 0.765 | 0.842 | 0.650 | 0.783 | 1.312 | 0.609 |
| PoseSOR (Guan & Lau, 2024a) | ECCV'24 | 0.732 | 0.561 | 0.619 | 2.355 | 0.664 | 0.750 | 0.542 | 0.670 | 1.925 | 0.547 |
| LG-SOR (Liu et al., 2025a) | CVPR'25 | 0.846 | 0.653 | 0.786 | 1.433 | **0.787** | 0.799 | 0.588 | 0.774 | 1.105 | **0.634** |
| Ours (One Stage) | - | 0.923 | 0.702 | 0.786 | 1.410 | 0.669 | 1.064 | 0.738 | 0.803 | 1.012 | 0.434 |
| Ours (Two Stage) | - | **0.932** | **0.710** | 0.788 | 1.413 | 0.676 | **1.075** | **0.743** | **0.824** | 0.978 | 0.431 |

trajectories collected from multiple human participants viewing natural scenes. For each image, we extract all fixation sequences from all observers and compute the transition probability between consecutive objects along each path, as:

$$P(s_t \mid I, s_{1:t-1}) = \frac{\text{Number of } \mathcal{S} \text{ with prefix } s_{1:t}}{\text{Number of } \mathcal{S} \text{ with prefix } s_{1:t-1}}. \quad (11)$$

Based on this, we construct the Attention Shift Probability Tree defined in Sec. 3.1. However, directly constructing trees from fixation sequences often includes many low-saliency objects, making the ASPT unnecessarily large and noisy and obscuring the main attention patterns. Hence, we compute for each object its visitation probability:

$$P_{\text{visit}}(A) = \frac{\text{Number of observers who fixate on } A}{\text{Total number of observers}}. \quad (12)$$

which measures the overall likelihood that object $A$ will be visited during viewing. We then prune nodes whose visitation probability falls below a threshold $\tau = 0.18$, which is empirically set based on the visitation probability distribution of salient objects in the ASSR dataset. Thus, we retain only objects with higher $P_{\text{visit}}$ as salient nodes in the tree.

Our final **ProbSOR-Bench** dataset comprises 15,000 probabilistic SOR samples, with 10,000 used for training and 5,000 reserved for testing. Each sample corresponds to one image with multiple attention-shift trajectories derived from human data. This benchmark comprehensively characterizes the stochasticity and diversity of human attention, providing a unified and reliable foundation for future research on probabilistic saliency ranking. See more statistical analysis in Appendix A of the Supplemental.

## 5. Experiments

### 5.1. Experimental Setup

**Dataset.** We evaluate our method against ten SOTA SOR models on two standard benchmarks, ASSR (Siris et al.,

2020) and IRSR (Liu et al., 2021a), to assess its generalization ability. We further compare it with the top four strongest baselines, (Liu et al., 2025a; Guan & Lau, 2024a; Wu et al., 2024; Deng et al., 2024), on ProbSOR-Bench to evaluate its performance under the probabilistic SOR setting. Detailed implementation and setting of our experiments can be found in Appendix Section B of the Supplemental.

**Metrics.** Previous metrics, such as SOR (Siris et al., 2020) and SA-SOR (Liu et al., 2021a), compute rank correlations (e.g., Spearman or Pearson coefficients) between the predicted and ground-truth rankings. Mathematically, these correlations measure the monotonic consistency between two orderings of the same set of objects. However, such formulations implicitly assume that there exists only *one* correct ranking, and any alternative ordering has to be treated as incorrect. For example, consider an image containing two salient objects, $o_a$ and $o_b$. If viewers exhibit two plausible fixation orders, $o_a \rightarrow o_b$ and $o_b \rightarrow o_a$, both are valid under a probabilistic interpretation of attention. Yet under SA-SOR, if the model predicts the alternative order, its correlation score becomes $\rho = -1$, even though the prediction remains semantically reasonable. Therefore, previous metrics fail to accommodate multiple plausible attention-shift patterns. In this work, we introduce **Probability-Aware Salient Object Ranking (PA-SOR)** as our probabilistic evaluation metric. Given a predicted attention-shift sequence (path) $\mathcal{S} = \{s_t\}_{t=1}^{T}$ for image $I$, we define its PA-SOR score directly as the path reward:

$$\text{PA-SOR}(\mathcal{S}) := R(\mathcal{S}) = \sum_{t=1}^{T} R(s_t), \quad (13)$$

where $R(s_t) = P(s_1, \ldots, s_t \mid I)$ is the step-wise reward in Eq. 3. Intuitively, PA-SOR($\mathcal{S}$) measures how plausible the predicted sequence is in terms of its likelihood. For the scores to be comparable across different images in the

dataset, we further compute a normalized PA-SOR:

$$\text{PA-SOR}_{Norm}(\mathcal{S}) = \frac{R(\mathcal{S})}{R(\mathcal{S}^*)}, \quad \mathcal{S}^* = \arg\max_{\mathcal{S}'} R(\mathcal{S}'). \quad (14)$$

Our reward formulation captures the *multi-path validity* of predicted sequences by assigning higher scores to paths that follow the human transition distribution. Further, we use $\text{IoU}_{mask}$, defined as the intersection-over-union between the predicted and GT masks, and MAE, defined as the mean absolute pixel difference between them, to assess salient-object localization accuracy. For fair comparison with prior works, we also report the SA-SOR scores on ASSR and IRSR using their original ground truths.

### 5.2. Comparison to Existing Methods

**Quantitative Results.** We first compare our method with ten SOTA SOR methods on the ASSR and IRSR benchmarks, as shown in Table 1. We report two variants of our model. *Ours (Stage 1)* denotes the model trained only with SFT, where we use the training set in ASSR or IRSR, and take the highest-reward path in our probabilistic annotation as the ground truth. Under this setting, our model already achieves the best PA-SOR and PA-SOR$_{Norm}$ scores on both ASSR and IRSR, while obtaining $\text{IoU}_{mask}$ and MAE that are comparable to the strongest existing methods. *Ours (Stage 2)* denotes the full ProbSOR training setting. Here, we train on the ProbSOR-Bench training set, which contains all training images from ASSR and IRSR as well as some additional images, while ensuring that no test images from ASSR or IRSR are included (Jiang et al., 2015). The model is optimized with our two-stage SFT+GRPO pipeline and then directly evaluated on ASSR or IRSR. This variant further improves our model's performance, demonstrating that learning from probabilistic supervision can be beneficial for predicting saliency ranks.

For a fair comparison and to assess generalization, we also measure SA-SOR using the GT rankings in ASSR and IRSR. On ASSR, our SA-SOR remains comparable to the best prior method, whereas on IRSR there is a noticeable drop. This gap mainly arises from the mismatch between the GTs of different benchmarks. Our annotations bias the model toward ProbSOR attention distribution, which is not fully aligned with the ASSR and IRSR labels. To further examine the gap between the GTs and metrics of the three benchmarks, we evaluate each benchmark's GTs under the evaluation setting of the other two benchmarks, as reported in Table 2. The results show that the deterministic labels of ASSR and IRSR are far from aligned. When evaluated with SA-SOR, the ASSR and IRSR ground truths only achieve about 0.5 with each other. Our probabilistic GTs attain a relatively high SA-SOR score on ASSR (0.765), but it is still not perfectly aligned, and its SA-SOR score on IRSR drops more noticeably to 0.496. In contrast, both ASSR and IRSR

*Table 2.* Cross-benchmark evaluation of ground truths. Each entry shows the score obtained when treating the column ground truth as prediction and evaluating it with the row metric. Yellow rows denote SA-SOR metrics, and the blue row denotes the PA-SOR$_{Norm}$ metric.

| Metric \ GT | ASSR GT | IRSR GT | ProbSOR GT |
|---|---|---|---|
| ASSR Metric | 1.0 | 0.593 | 0.765 |
| IRSR Metric | 0.525 | 1.0 | 0.496 |
| ProbSOR Metric | 0.728 | 0.511 | 1.0 |
| ProbSOR Metric | 0.745 | 0.621 | 1.0 |

*Table 3.* Performance of state-of-the-art SOR methods on ProbSOR-Bench. Best results are marked in bold and second-best results are underlined.

| Method | Reference | PA-SOR ↑ | PA-SOR$_{Norm}$ ↑ | IoU$_{mask}$ ↑ | MAE ↓ |
|---|---|---|---|---|---|
| QAGNet | CVPR'24 | 0.829 | 0.610 | **0.789** | **1.131** |
| DSGNN | CVPR'24 | 0.857 | 0.667 | 0.781 | 1.420 |
| PoseSOR | ECCV'24 | 0.731 | 0.572 | 0.648 | 2.173 |
| LG-SOR | CVPR'25 | 0.826 | 0.604 | 0.770 | 1.276 |
| Ours | - | **0.955** | **0.711** | 0.766 | 1.504 |

ground truths obtain relatively high scores under our Prob-SOR PA-SOR$_{Norm}$ metric (0.745 and 0.621, respectively). This is because PA-SOR$_{Norm}$ explicitly accounts for multiple plausible attention paths. So, many sequences that are penalized as "incorrect" under ASSR/IRSR's single-path SA-SOR metrics are considered as reasonable and receive high scores under our probabilistic evaluation. For completeness, we further evaluate four SOTA SOR methods with the highest SA-SOR scores on the ASSR, and report their results on our ProbSOR-Bench, as shown in Table 3.

**Qualitative Results.** Figure 3 shows qualitative comparisons on several scenes from ProbSOR-Bench. For each image, we visualize the Top-1 to Top-4 attention paths from ASPT, followed by the predictions of four latest methods and ours, together with their PA-SOR and SA-SOR scores. Existing SOR models often miss fine-scale foreground instances (first row) or leak saliency into the background (second row), leading to incorrect rankings. In contrast, our method produces clean instance masks and ranking orders that are visually more consistent with the multi-path ground-truth trajectories. Beyond the visual differences, the PA-SOR and SA-SOR scores also highlight the limitation of previous deterministic metrics. In the second example, one baseline attains a reasonably high PA-SOR (0.64) but a strongly negative SA-SOR (-0.60), showing that SA-SOR severely penalizes predictions that merely permute a few plausible objects, while PA-SOR still regards them as moderately consistent with the probabilistic ground truth.

### 5.3. Ablation Analysis

**Effect of Two-stage Optimization.** We first investigate the contribution of each training stage by comparing four

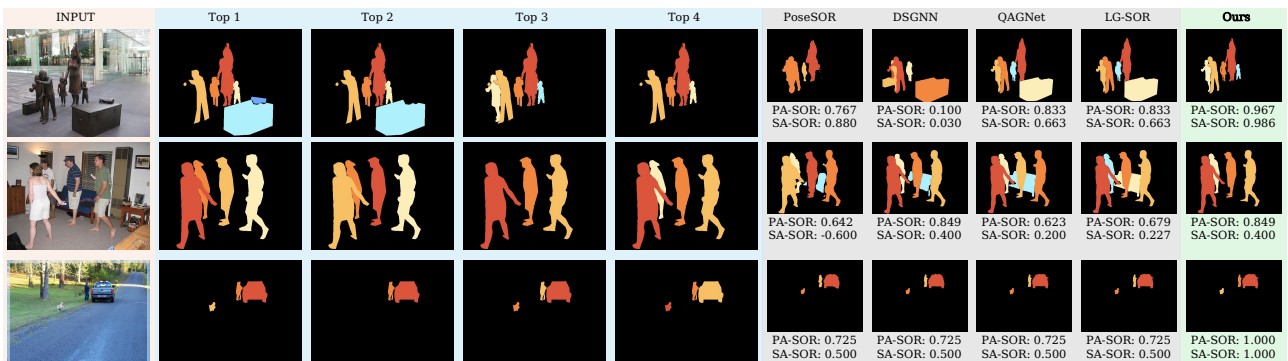

*Figure 3.* **Qualitative comparison.** For each example, we show the input image, the Top-1–Top-4 probable attention paths from the ASPT, the predictions of the four best-performing SOR methods on ASSR, and our ProbSOR prediction. PA-SOR and SA-SOR scores are reported for each method to quantify its discrepancy with the probabilistic and deterministic GTs.

variants of our model: (1) the vanilla Qwen3-VL, (2) Baseline + SFT, (3) Baseline + GRPO, and (4) the full training pipeline with both SFT and GRPO. As shown in Table 4, Baseline+SFT already brings consistent gains across all metrics, showing that learning from the highest-reward attention paths provides effective supervision. Applying GRPO alone brings only limited gains, since the frozen model has not yet learned the structured output format or the notion of attention paths, and thus ASPT-sampled candidates are largely suboptimal from the model's perspective, and the resulting group-wise advantages offer weak and noisy signals that are insufficient to drive policy updates. The best results are achieved when combining SFT with GRPO, indicating that our two-stage training pipeline is crucial for fully capturing probabilistic attention-shift behaviors.

*Table 4.* Ablation study on the training pipeline. We compare the effects of SFT and GRPO optimization.

| Method | PA-SOR ↑ | PA-SOR$_{Norm}$ ↑ | IoU ↑ |
|---|---|---|---|
| Baseline (Qwen3-VL) | 0.657 | 0.478 | 0.618 |
| + SFT | 0.934 | 0.697 | 0.740 |
| + GRPO | 0.671 | 0.493 | 0.637 |
| + SFT + GRPO (Ours) | **0.945** | **0.708** | **0.771** |

**Reward and GRPO Analysis.** We further investigate the impact of the GRPO sampling strategy on model performance, as reported in Table 5. The Baseline corresponds to the model after Stage-1 SFT. Setting I adopts the standard GRPO scheme, in which the current policy samples G candidate sequences online for each group. Setting II employs the proposed offline sampling scheme, in which G candidate paths per image are selected from the ASPT; configurations with G = 4, 8, and 12 are evaluated. The results show that in Setting I, online sampling from the policy slightly degrades some metrics compared with the SFT baseline. This suggests that, when the policy is not yet fully aligned with the task, the sampled groups often fail to contain a sufficient

number of high-reward and diverse paths, leading to noisy advantage estimates and unstable update signals. In contrast, the offline ASPT sampling in Setting II produces clear performance gains: increasing G from 4 to 8 improves PA-SOR and PA-SOR$_{Norm}$, while further increasing to G=12 yields similar performance with diminishing returns. These observations indicate that, provided each group contains an adequately rich set of high-reward candidates, a moderate sampling budget is sufficient to effectively guide the policy toward a more reasonable distribution over attention paths.

*Table 5.* Ablation on the GRPO sampling strategy on ProbSOR-Bench. The Baseline corresponds to the Stage-1 SFT model. Setting I uses standard GRPO with online sampling from the current policy. Setting II uses the proposed offline sampling scheme with $G$ candidates drawn from the ASPT.

| Method | Setting | Sampling | $G$ | PA-SOR ↑ | PA-SOR$_{Norm}$ ↑ | IoU$_{mask}$ ↑ |
|---|---|---|---|---|---|---|
| Baseline (SFT) | – | – | – | 0.934 | 0.697 | 0.740 |
| + GRPO | I | Policy | 4 | 0.921 | 0.681 | 0.732 |
| | | | 8 | 0.926 | 0.686 | 0.738 |
| | | | 12 | 0.923 | 0.685 | 0.731 |
| + GRPO | II | ASPT | 4 | 0.939 | 0.703 | 0.770 |
| | | | 8 | 0.944 | 0.708 | 0.766 |
| | | | 12 | 0.942 | 0.708 | 0.767 |

**Box-only Validation.** Since our framework uses a frozen SAM 2 decoder to convert predicted boxes into instance masks, we conduct a box-only validation to isolate the model's ranking and localization ability from the potential influence of mask decoding quality. Specifically, for QAG-Net, which directly predicts instance masks, we convert each predicted mask into its enclosing bounding box without changing the model architecture. For our method, we directly use the bounding boxes generated by the VLM. We then evaluate the predicted object order with PA-SOR$_{Norm}$ and measure localization quality with Box-IoU. The results show that our method achieves a higher PA-SOR$_{Norm}$ than QAGNet (0.710 vs. 0.652), while maintaining competitive

box localization quality (0.774 vs. 0.827). This indicates that the improvement in probabilistic ranking mainly comes from modeling human attention-shift distributions, rather than from the SAM 2 mask decoder.

**Effect of VLM backbone.** We further study whether the proposed probabilistic SOR framework depends on a specific VLM backbone. To this end, we replace the backbone with several open-weight VLMs and keep the same Stage-1 training setting, where the model is supervised by the highest-reward path from the ASPT. Table 6 shows that different VLM backbones achieve reasonable performance under the proposed formulation, while stronger backbones generally lead to better ranking and localization performance. In particular, Qwen3-VL-4B achieves the best performance among the tested lightweight backbones, with PA-SOR$_{\text{Norm}}$ of 0.693 and BBox-IoU of 0.774. These results suggest that ProbSOR is not tied to a single backbone, although the final performance is still influenced by the visual reasoning and localization capability of the underlying VLM.

*Table 6.* Effect of different VLM backbones under the same Stage-1 training setting.

| Backbone | PA-SOR$_{\text{Norm}}$ ↑ | BBox-IoU↑ |
|---|---|---|
| LLaVA-OneVision-1.5-4B | 0.636 | 0.621 |
| Qwen2.5-VL-3B | 0.679 | 0.771 |
| Qwen3-VL-4B | 0.693 | 0.774 |

## 6. Limitation

Our method remains challenged in crowded scenes where multiple similar instances overlap or heavily occlude each other (e.g., people in a crowd). Under strong overlap, instance boundaries become ambiguous and the predicted masks may partially merge or shift across adjacent instances. As a result, although the model often identifies the correct salient category or region, it may assign the highest rank to a nearby instance when object boundaries are ambiguous.

## 7. Conclusion

In this work, we revisited salient object ranking through a probabilistic lens and showed that human attention is better modeled as a distribution over multiple plausible object orders rather than a single deterministic sequence. Building on this insight, we introduced ProbSOR, which encodes attention-shift behavior as an Attention-Shift Probability Tree and learns a reward-weighted distribution over paths with a two-stage training pipeline. We further proposed the PA-SOR and PA-SOR$_{Norm}$ metrics and constructed ProbSOR-Bench from human fixation trajectories, enabling principled training and evaluation under probabilistic super-

vision. Extensive experiments demonstrate that our model achieves strong performance on conventional ASSR/IRSR benchmarks and ProbSOR-Bench.

## Acknowledgements

This work is in part supported by a GRF grant from the Research Grants Council of Hong Kong (Ref: 11220724).

## Impact Statement

This work advances salient object ranking by modeling human attention shifts as a probabilistic distribution rather than a single deterministic sequence. The proposed framework, benchmark, and metrics may benefit research on visual saliency, human-centered scene understanding, image interpretation, and vision systems that need to reason about multiple plausible object-attention paths in complex scenes. More broadly, this work encourages evaluation protocols that better reflect the diversity of human perception. As with other data-driven vision methods, the learned distributions may depend on the coverage and annotation protocol of the underlying fixation data, and future work may study their generalization across domains, scene types, and observer groups. This work has no associated risks to security, privacy, or fairness.

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

# A. Dataset

To examine how many attention paths are associated with each image, we first count the number of images that contain a given number of valid paths. As shown in Figure 4, more than half of the images have between 1 and 20 paths. This indicates that, even for the same scene, observers often follow many different yet plausible attention sequences rather than a single "correct" order. In addition, the distribution exhibits a clear long tail, and for some complex scenes the number of paths exceeds 60, revealing substantial diversity in human attention patterns.

We further analyze how much probability mass is captured by the most likely paths. For each image with $K$ paths, we sort the paths in descending order of probability and compute the cumulative probability of the top-$k$ paths($k = 1, \ldots, K$). The boxplots in Figure 4 summarize the distribution of this top-$k$ coverage over all images, with the red dashed line indicating the median. Typically, the top-1 path already accounts for about one third of the total probability, and the top 5–10 paths cover roughly 70–80%. When k increases to around 30, the mean coverage approaches 90%. In contrast to conventional SOR benchmarks, which provide only a single ground-truth path per image, these results clearly show that human attention is inherently stochastic and cannot be faithfully represented by a unique ranking. Instead, ProbSOR-Bench offers multiple attention-shift patterns based on probability, providing a rich and realistic basis for modeling the full distribution of human attention trajectories.

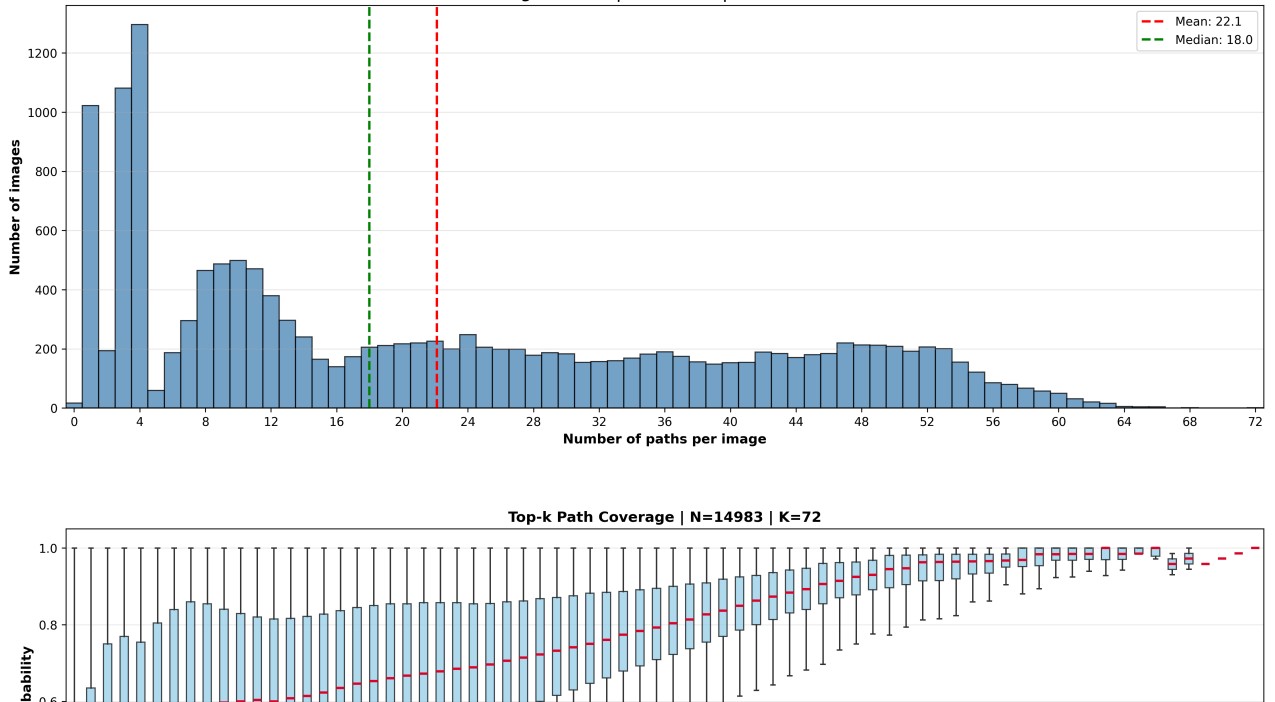

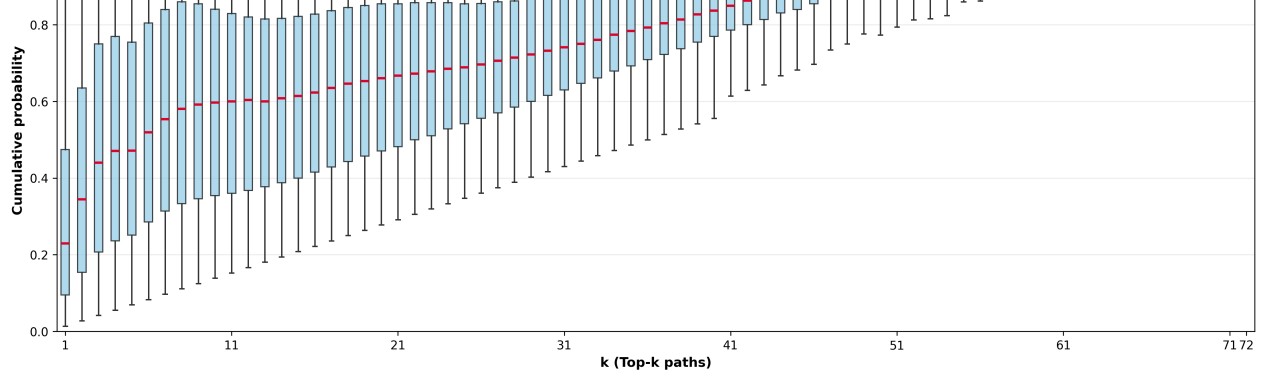

*Figure 4.* **Statistics of attention paths in ProbSOR-Bench.** Top: Histogram of the number of valid attention paths per image, with most images having 1–20 paths and a long tail beyond 60 for complex scenes. Bottom: Cumulative probability covered by the top-$k$ most likely paths. For each $k$, the boxplot summarizes the distribution across images and the red dashed line denotes the median.

# B. Implementation Details.

We use Qwen3-VL (8B) as the VLM backbone and apply LoRA adapters for efficient task adaptation. During Stage-1 Supervised Fine-Tuning, the model is trained for 2 epochs (2,500 optimization steps) using a per-device batch size of 1 with 8-step gradient accumulation. SFT supervision is constructed from the highest-reward attention path. We use the fused AdamW optimizer with a learning rate of $2 \times 10^{-5}$, a constant schedule with warm-up, and gradient clipping. Images are resized to 448×448. For Stage-2 GRPO optimization, we initialize from the SFT checkpoint and, for each image, sample $G = 8$ candidate sequences from the ASPT together with their rewards. We discard images with fewer than three attention paths, as they provide limited diversity and contribute little to GRPO optimization. The GRPO stage is also trained for 2 epochs, using a clipping coefficient of $\epsilon = 0.2$ and a mild KL penalty ($\beta = 0.1$) to stabilize policy updates. Only the LoRA-adapted language-model parameters are updated in this stage, and we use the AdamW optimizer with a learning rate of $5 \times 10^{-6}$. During inference, we use sampling-based decoding with temperature $0.7$ and nucleus sampling with top-$p = 0.95$, allowing the model to generate diverse plausible attention-shift paths for the same image. For mask extraction, we employ a frozen SAM 2 model to obtain instance masks, using the predicted bounding boxes as box prompts. All experiments are conducted on 4 NVIDIA A100 GPUs.

