# OpenReview forum: "Probabilistic Salient Object Ranking"
_ICML.cc/2026/Conference — ICML 2026 regular_

### Official Review · Reviewer_199B · 2026-02-28

**Soundness:** 3
**Presentation:** 3
**Significance:** 3
**Originality:** 3
**Overall Recommendation:** 4
**Confidence:** 4

**Summary:**

This paper raises an important issue in the salient object ranking (SOR) task and existing solutions that there is a mismatch between “human stochastic shifted attention” and “the only one fixed ground truth”. Based on the above observation, this paper addresses SOR in a probabilistic perspective that not evaluate the ranking result by single fixation trajectory, but a probability distribution.

To meet the requirements of the aforementioned tasks formulation, this paper constructs a new benchmark **ProbSOR-Bench** and a new metric **PA-SOR** (along with its normalized version). VLM and GRPO are adopted to accomplish this task.

**Compliance With Llm Reviewing Policy:**

Affirmed.

**Final Justification:**

After reviewing the author rebuttal and other reviewers' comment, I find most of my concerns have been addressed. I choose to maintain the initial recommendation according to the novelty and quality of the paper, and raise my confidence level slightly.

**Key Questions For Authors:**

Please try to address the following concerns, and I will determine the final score based on the author's responses and the comments from other reviewers.

**1.** In *[Page 5 Line 232-234]*, the authors state that the “category” information is also input to the SAM, however, as far as I know, SAM2 does not support “class” prompt. Since the authors claim that they adopted SAM2 is frozen, how did they address this issue?

**2.** According to Table 1, none of the compared method is foundation model-based (except that LG-SOR adopted VLLM to obtain an image caption for reasoning). The proposed method in this paper use:
- a VLM to reason “rank”, “category”, and “bbox” of the candidate salient objects ( these factors are directly correlated with the quality of the final performance evaluation).
- a SAM2 to generate the segmentation mask (which is also directly correlated to IoU, MAE, and SA-SOR metrics)

All the above may lead to a potential unfair comparison, that the performance boost stems not from the new task formulation itself, but rather from the powerful reasoning capabilities of the foundation models.

**3.** Under the SFT setting (one stage training), the proposed method achieves a relatively low SA-SOR (compared with QAGNet, DSGNN, and LG-SOR), why does this happen? Is it due to the unpleasant segmentation mask, or sub-optimal predicted order? Still in the SFT setting, why the proposed method achieve the best PA-SOR and normalized PA-SOR, since “probability” and “reward” are not introduced in this stage yet. What is the underlying relationship between PA-SOR and SA-SOR? Are these two indicators mutually conflicting?

**4.** Compared to the SA-SOR metrics, although the proposed PA-SOR considers all plausible attention transition trajectories in a probabilistic manner, it lacks the evaluation on segmentation mask, which is also a critical aspect in salient object detection (SOD) and SOR. I wonder if the segmentation mask evaluation is taken into consideration in the Reward and PA-SOR Metric, will the final result be different?

**5.** Since the proposed method leverages the power of VLM, what is the impact of different VLMs (i.e., LLaVa, OPERA, GPT-4V for example in LG-SOR) to the result? And the computational efficiency comparison between other methods is also needed.

**Limitations:**

Although the limitations are discussed in the *Appendix*, I recommend the author add this part into the main content of the paper with more visual results.

**Strengths And Weaknesses:**

**Strengths:**
- The motivation is quite insightful. The observation and raised issue on existing SOR task and solutions is very important.
- The paper is well-structured, with clear presentation and definition to new task formulation.
- The experimental results are sound.

**Weakness:**
- Some details in the methodology and experiments are missing.
- Potential unfair comparison with existing solutions.

---

> ### Author Rebuttal · Authors · 2026-03-31
>
> Q1: Thank you for catching this. We agree that the current wording in Eq. 7 is inaccurate. The frozen SAM 2 model uses only the image and the predicted bounding box as prompts to produce the mask. The category token is generated by the VLM as part of the structured output sequence to describe the semantic identity of each ranked object.
>
> ---
>
> Q2: We appreciate this concern, but our improvements are not simply due to a VLM and SAM 2.
>
> First, our method is explicitly decoupled. The VLM predicts the salient-object order and bounding boxes, while SAM 2 is used only afterward as a frozen mask decoder. Thus, the main ranking logic is determined by the VLM and the probabilistic training objective, whereas SAM 2 only converts predicted boxes into masks.
>
> Second, if the gain mainly came from a trivial VLM/SAM 2 advantage, we would expect our method to dominate all related metrics. However, this is not what Tables 1 and 3 show. Our method is not uniformly best on $IoU_{mask}$ and $MAE$, and previous methods such as QAGNet and LG-SOR still outperform us on some mask-related results. Moreover, in a box-only setting without SAM 2, our method still maintains a clear $\text{PA-SOR}_{Norm}$ advantage over QAGNet (0.710 vs. 0.652) with competitive Box-IoU (0.774 vs. 0.827). This indicates that the gain mainly comes from better modeling of human attention transitions rather than from the mask decoder.
>
> Third, the benefit of our method is also visible when the backbone is fixed. In Table 4, using the same Qwen3-VL backbone, Baseline + SFT already improves the reward from 0.657 to 0.934, and SFT + GRPO further improves it to 0.945. This indicates that the gain is not reducible to model scale alone, but also comes from probabilistic supervision and two-stage optimization.
>
> Overall, our main contribution is the probabilistic reformulation of SOR and the probability-aware learning/evaluation framework. We agree that model capacity is relevant and will clarify this more explicitly in the revision.
>
> ---
>
> Q3: The lower SA-SOR under SFT-only mainly comes from the mismatch between probabilistic supervision and deterministic GT rankings, rather than simply poor masks. In Stage 1, the model is already supervised by the highest-reward path from the probabilistic annotation, not by the original single-path GT of ASSR/IRSR. Therefore, it is already biased toward the ProbSOR attention distribution, which explains why SFT alone can achieve the best PA-SOR even before GRPO. Our ablation confirms that Baseline + SFT already yields large gains (see Table 4), while GRPO further improves the result by learning from multiple plausible paths and their relative rewards.
>
> This also explains why PA-SOR is necessary. SA-SOR assumes one fixed target ranking and can penalize alternative but human-plausible paths. In contrast, PA-SOR is designed to evaluate whether a predicted path is plausible under the human attention distribution. Please refer to `reviewer@k6Dh W2/Q2` for explicit explanation about our metric PA-SOR.
>
> ---
>
> Q4: We agree that segmentation quality is important in SOD/SOR. However, PA-SOR is intentionally designed as a path-level metric to evaluate the plausibility of attention-shift sequences under human viewing behavior, while mask quality is evaluated separately by $\text{IoU}_{mask}$ and MAE.
>
> If mask quality were incorporated, the metric would become a joint ranking-and-segmentation objective, requiring fair weighting between the two and making it harder to distinguish whether gains come from better attention modeling or better mask decoding.
>
> Therefore, we evaluate attention-transition ordering with PA-SOR and segmentation with $\text{IoU}_{mask}$/MAE separately, so that probabilistic attention modeling can be evaluated without being confounded by the mask decoder.
>
> ---
>
> Q5: First, our framework is not tied to a single VLM. Under the same Stage-1 setup, we additionally tested several open-weight backbones. We will include the following results in Table 3: Qwen3-VL-4B, Qwen2.5-VL-3B, and LLaVA-OneVision-1.5-4B achieve $\text{PA-SOR}_{Norm}$ of 0.693, 0.679, and 0.636, respectively, with corresponding bbox-IoU of 0.774, 0.771, and 0.621. These results suggest that our framework is not specific to a single backbone, while also indicating that backbone capability affects the final quality.
>
> Second, regarding efficiency, our Qwen3-VL-8B takes 571ms/image (512×512) versus QAGNet’s 307ms (768×768). However, our method retains a practical deployment advantage: it uses a standard Transformer-based VLM without task-specific operators or custom graph modules, making it compatible with mature inference stacks such as vLLM and SGLang and thus amenable to substantial batching and throughput improvements. More importantly, our main claim is not that a larger VLM alone explains the gains, but that we reformulate SOR as a probabilistic framework (ProbSOR) and propose a new dataset, benchmark, and metric (PA-SOR).
>
> ---
>
> L1:Agree. Will do.

---

> > ### Author Rebuttal · Reviewer_199B · 2026-04-01
> >
> > I acknowledge the authors’ efforts during the rebuttal period. Most of my major concerns have been addressed.
> >
> > This paper is well-motivated, and the core idea is insightful, **with clear potential to advance research in the SOR field**.
> > However, **several details, particularly regarding the experimental design and the proposed metric, remain insufficiently clarified**.
> >
> > Therefore, **I maintain my recommendation as [4-Weak Accept] with moderate confidence**.
> >
> > ### **Additional comments:**
> > **(_No more additional experiments are needed, and addressing the following points may not necessarily increase my overall score, but could influence my confidence level._)**
> >
> > **1.** In the rebuttal to **[Reviewer Teqa Q1/W1 3.]**, the authors mention a "Box-Only Validation." I would appreciate further clarification on how this experiment is conducted. To my knowledge, the original QAGNet directly produces segmentation results rather than bounding boxes. Does this imply that the architecture of QAGNet was modified for this experiment? Additionally, on which benchmark (ASSR, IRSR, or ProbSOR-Bench) is this validation performed?
> >
> > **2.** After cross-referencing **Reviewer Teqa’s comments**, I share the concern regarding potential **"Metric Circularity"** in the GRPO training process. However, as I do not have deep expertise in LLM/VLM training protocols, **I defer to Reviewer Teqa’s assessment on the validity of the VLM utilization and training strategy.**
> >
> > **3.** I am still unclear about my previous Q3. The authors attribute the performance gap observed during the SFT stage to "biased supervision." I note from Table 2 that the ground truth annotations for the same image may differ across ASSR, IRSR, and the proposed ProbSOR-Bench. This raises two questions (I also recommend **adding a clarification on the following issues in Sec. 5.1 or Sec. 5.2**.):
> > - (i) *Is there any overlap among samples across these datasets?* and
> > - (ii) *What criteria or annotation protocols were used to define the ground truth in each dataset?*

---

> > > ### Author Response · Authors · 2026-04-08
> > >
> > > We sincerely thank the reviewer for the positive assessment of the paper’s motivation and potential impact. We are also glad that our rebuttal has addressed most of the reviewer’s major concerns, and we appreciate the opportunity to further clarify the remaining points below.
> > >
> > > ---
> > >
> > >
> > > **1. About the ''Box-Only Validaton'':**
> > >
> > > The purpose of this experiment is to remove the effect of the SAM 2. For QAGNet, the box is obtained from its predicted salient instances by converting each predicted instance mask into its enclosing bounding box, and there is no architectural modification. Our method directly uses the VLM-predicted boxes. The localization quality is reported with Box-IoU. This experiment is performed on the ASSR benchmark setting.
> > >
> > > ---
> > >
> > >
> > > **2. About the concern of ''Metric Circularity'':**
> > >
> > > Using this reward for GRPO training is justified for two reasons. First, Stage 2 is introduced to model the distribution of plausible attention shifts. Otherwise, the model would collapse to fitting only a single GT path, as in previous methods. The reward is derived from human attention data and serves as a natural signal for GRPO to encourage the model to assign higher probability to multiple high-reward plausible paths. Table 1 shows that Stage 2 further improves PA-SOR over Stage 1. Moreover, after removing the best path from 10 sampled paths on 200 test images, the post-GRPO model outperforms the pre-GRPO baseline (0.611 vs. 0.592), confirming that Stage 2 improves modeling of the attention-shift distribution rather than merely reinforcing the top path. Second, this training does not involve any leakage of test data, GT annotations, or test rewards. Our model uses reward signals constructed only from the training set to learn the distribution of plausible human attention shifts.
> > >
> > > ---
> > >
> > > **3.About the performance gap, dataset overlap, and GT definitions:**
> > >
> > > GT annotations for the same image differ across ASSR, IRSR, and ProbSOR-Bench. Therefore, directly evaluating our model trained on ProbSOR-Bench under the ASSR/IRSR deterministic protocols can naturally lead to performance drops due to supervision-target mismatch. More clarification on these datasets is provided below.
> > >
> > > 1. Overlap among datasets:
> > >
> > > All of these benchmarks are derived from SALICON images. We use the full SALICON split, i.e., 10,000 images for training and 5,000 images for testing, whereas ASSR and IRSR are constructed from subsets of SALICON and are therefore also subsets of our benchmark. Specifically, ASSR contains 7,646 of our 10,000 training images and 2,418 of our 5,000 test images, while IRSR contains 6,059 of our 10,000 training images and 2,929 of our 5,000 test images. We compute the cross-metric results using only the corresponding subset images shared by each benchmark pair.
> > >
> > > 2. Ground-truth criteria:
> > >
> > > **ASSR** defines GT with DistFixSeq, which reduces SALICON fixation sequences to a single deterministic ranking by ordering distinct fixated objects and aggregating their scores across observers. **IRSR** defines GT by ranking salient instances according to the maximum SALICON saliency value within each instance mask. For **ProbSOR-Bench**, we construct the ASPT by grouping trajectories with shared prefixes and estimating transition probabilities from their empirical frequencies. Based on these probabilities, we define the reward for each path, so that the GT is represented as a distribution over plausible attention-shift paths rather than a single deterministic ranking.
> > >
> > > We will add this clarification in revision.
> > >
> > > ---
> > >
> > > We thank the reviewer again for the constructive suggestions. We hope that our responses help resolve the remaining concerns, and we would greatly appreciate your positive consideration.

---

### Official Review · Reviewer_k6Dh · 2026-03-11

**Soundness:** 2
**Presentation:** 2
**Significance:** 3
**Originality:** 3
**Overall Recommendation:** 3
**Confidence:** 4

**Summary:**

Existing Salient Object Ranking (SOR) methods model salient object ranking in a deterministic manner. However, human attention shifts are inherently stochastic and variable; current methods and evaluation metrics fail to account for this randomness. This paper proposes a Vision-Language Model (VLM)-based probabilistic SOR framework that learns the uncertainty of attention shifts via GRPO. Furthermore, it introduces a probability-aware evaluation metric that measures the rationality of a sequence by accumulating transition probability rewards along the path, thereby providing a more reliable and fair evaluation. Additionally, a new dataset is constructed.

**Compliance With Llm Reviewing Policy:**

Affirmed.

**Key Questions For Authors:**

1.	The construction of the ASPT is not described in sufficient detail, making it confusing to follow. Line 206 mentions that "the final attended objects are linked to a terminal leaf node." However, why is only S2 followed by a leaf node? Additionally, in the figure, the values of P and R in the second row of the ASPT do not align. According to Equation 3, shouldn't P and R be consistent?

2.	The paper proposes a metric named PA-SOR, which is defined using the path rewards utilized in the text. This metric seems exclusive to this paper, as other methods do not feature such path rewards. Furthermore, based on the description, the calculation of these rewards does not appear to incorporate ground truth values. Is this evaluation mechanism fair? The proposed method seems to achieve state-of-the-art (SOTA) performance only under its own proposed metric, while underperforming compared to alternative methods on other metrics.

3.	Does 'P' refer to the same variable in Equation 3 and Equation 11? Rewards should logically be defined based on predictions, yet 'P' in Equation 11 appears to denote the ground truth. This is somewhat confusing and requires clarification.

**Limitations:**

Yes

**Strengths And Weaknesses:**

Strengths
The paper rethinks current SOR methods from the perspective of the diverse patterns in human attention shifts. It proposes a more realistic task from a novel viewpoint, along with a corresponding methodology and dataset.
Weaknesses
1.Certain descriptions in the text lack sufficient detail, such as the predictive outputs of each sub-model, the construction of the ASPT, and the explanations of specific formulas.
2.The definition of the proposed PA-SOR metric appears to be problematic; please refer to the "Key Questions" section for details.

---

> ### Author Rebuttal · Authors · 2026-03-31
>
> We thank Reviewer k6Dh for the constructive feedback. We appreciate the recognition of the novelty of reformulating SOR from a probabilistic perspective, as well as the value of the new task setting and dataset. Below we respond point-by-point.
>
> ---
>
> W1/Q1: We will revise the paper to improve clarity.
>
> First, regarding the outputs of each sub-module: the VLM outputs a structured ranking sequence in the format $\{|rank_t|category_t|bbox_t|\}_{t=1}^T$, i.e., an ordered list of salient objects together with category labels and bounding boxes (Eq. 5–6). SAM 2 is then used only as a mask decoder to convert each predicted $bbox_t$ into a pixel mask $M_t$ (Eq. 7).
>
> Second, regarding ASPT construction: for each image, we collect fixation trajectories from all observers (about 60 observers), map each fixation to its corresponding salient instance, and convert each trajectory into an object-level sequence. Conditional transition probabilities are then estimated by frequency counting over sequence prefixes (Eq. 11), which define the ASPT edges (L206–L208).
>
> Third, we agree that the notation around probabilities and rewards can be clearer. In particular, after re-checking the equations and Figure 2(b), we found Eq. 3 contains a typo. The intended step-wise reward is the probability of the visited prefix, not the local conditional transition alone, i.e., $R(s_t) = P(\{s_1, \ldots, s_t\} \mid I)$. This is also consistent with the values shown in Figure 2(b), e.g., the reward at $S_2$ along the path $R → S_3 → S_2$ is $0.165 = 0.55 \cdot 0.3$.
>
> Q1: We omit the leaf node in Figure 2(b) for illustrative simplicity. In the actual ASPT, every complete attention-shift path terminates at a leaf. As for $P$ and $R$, $P$ denotes either the probability of a path or the conditional probability of shifting from one object to another, whereas $R$ denotes the reward, which in our formulation is the probability of the path prefix up to the current node, i.e., $R(s_t) = P(\{s_1, \ldots, s_t\} \mid I)$.
>
> We will clarify these points and correct Eq. 3 accordingly in the revision.
>
> ---
>
> W2/Q2: It is important to emphasize that PA-SOR is a new metric proposed to reconcile deterministic SOR with the inherent stochasticity of human attention. We strongly disagree with the characterization that this metric is unfair or tailored solely to our method. Instead, it serves as a general measure of how plausible a predicted attention shift path is, which is also our motivation for proposing PA-SOR.
>
> 1. Limitation of previous metrics: Previous SOR metrics, such as SA-SOR assume a single correct ranking, whereas human attention shifts exhibit variability and stochasticity. Under a single-path metric, an alternative but human-plausible sequence may receive an overly harsh penalty (L305-306). Instead, PA-SOR serves as a generalizable metric capable of evaluating diverse human-plausible sequences by assigning scores based on their alignment with the empirical distribution of human attention shifts (see Figure 1). We believe this is a better fit for the probabilistic SOR setting studied in this paper.
>
> 2. Fairness of PA-SOR:  PA-SOR is computed entirely from the human annotation distribution encoded in the ASPT. It does not rely on any reward predicted by our model. Any competing method can be evaluated under PA-SOR by simply taking its predicted ranking sequence and querying the same human-derived ASPT. In this sense, PA-SOR is a dataset-level evaluation protocol, not a method-dependent scoring rule. Additionally, it explicitly accommodates multiple plausible sequences, addressing a key limitation of prior deterministic SOR metrics.
>
> 3. Reason for the performance gap:  Lower PA-SOR scores of prior methods mainly stem from their different learning objective: matching a single GT ranking rather than modeling the distribution of plausible human attention shifts. Including PA-SOR in Table 1 is not meant only to show that our method performs well under PA-SOR, but also to make explicit the gap between single-GT supervision and the inherently probabilistic nature of human attention. For completeness and compatibility with prior work, we also report the SA-SOR results on ASSR and IRSR (Table 1). Our method may underperform on SA-SOR in some cases because paths that are reasonable under human attention are penalized by deterministic evaluation. Table 2 further shows this gap across benchmarks: even using our GT as prediction gives only SA-SOR 0.765 on ASSR and 0.496 on IRSR, indicating an approximate upper bound under cross-benchmark deterministic evaluation. Therefore, our scores of 0.676 and 0.431 are acceptable.
>
> ---
>
> Q3: As mentioned in our response to W1/Q1, we will correct Eq. 3, which computes the step-wise reward. Please note that our rewards are computed directly from the human-derived ASPT using a rule-based strategy, without relying on any learned reward/value model. Distinctly, Eq. 11 defines the transition probability used during ASPT construction.

---

> > ### Author Rebuttal · Reviewer_k6Dh · 2026-04-03
> >
> > I have carefully read the authors' rebuttal, as well as the comments and discussions from the other reviewers. I appreciate the authors' rebuttal, particularly the clarifications regarding the sub-modules' predictive outputs, the ASPT construction process, and the correction of the typo in Equation 3. The manuscript tackles an interesting problem，shifting the modeling and evaluation of Salient Object Ranking (SOR) from deterministic sequences to probabilistic distributions to better capture human visual attention shifts.Despite these clarifications, however, my fundamental concerns regarding the fairness of the evaluation and the formulation of the proposed metric remain somewhat unresolved. My main reasons are detailed below:
> > 1. The authors maintain that PA-SOR is a fair dataset-level protocol based on human annotations rather than a learned reward. However, the core issue of circularity persists. During GRPO, the model is explicitly trained to maximize rewards drawn directly from the offline ASPT. The PA-SOR evaluation metric then mathematically accumulates these exact same path rewards. Testing the model on its own training objective makes the comparison against standard baselines fundamentally unfair. The model's SOTA performance on its own metric demonstrates that it successfully fitted the training reward function, rather than conclusively proving superior generalization in attention modeling.
> > 2. The proposed method underperforms compared to baselines when evaluated on established, standard metrics (e.g., SA-SOR on IRSR). The authors attribute this to a ground-truth mismatch, mentioning the model is inherently biased toward the ProbSOR distribution. But, I take this as a major limitation. If the model only excels on a self-defined metric and dataset (PA-SOR) but drops in performance on standard deterministic benchmarks, it raises serious questions about the robustness.
> > 3. The rebuttal clarified that the reward R in Equation 3 (correcting the typo) is simply the probability of the visited prefix based on dataset frequencies. Because the model relies on offline sampling directly from the ASPT rather than active policy exploration, Stage 2 training functionally reduces to a weighted supervised learning or contrastive optimization over static dataset statistics. The method is essentially supervised preference alignment, and framing it as Reinforcement Learning (GRPO) somewhat overstates its novelty.
> > Therefore, I am inclined to maintain my original score.

---

> > > ### Author Response · Authors · 2026-04-08
> > >
> > > We thank the reviewer for the follow-up and for acknowledging our clarifications regarding the predictive outputs, ASPT construction, and Eq. 3. Below we further clarify the reviewer's continued concerns.
> > >
> > > ---
> > >
> > > Q1:
> > > We would like to clarify that PA-SOR is the evaluation metric under our newly introduced probabilistic SOR setting, and is designed to measure how well a model captures the distribution of human attention shifts. In contrast, prior SOR methods were designed to fit a single deterministic GT path. Their datasets, objectives, and algorithms do not explicitly model attention distributions or multiple plausible paths. Therefore, they CANNOT be directly optimized for PA-SOR. Evaluating prior methods under PA-SOR is intended to reveal the gap between deterministic GT supervision and modeling the distribution of human attention shifts. Their relatively low PA-SOR scores indicate that such models do not adequately capture the distribution of plausible human attention-shift paths.
> > >
> > > Within our training framework, Stage 2 GRPO is designed to move beyond supervision on only the best path and encourage the model to capture a broader distribution of plausible attention shifts. Otherwise, training with only Stage 1 would collapse back to deterministic single-GT fitting. To verify this, we sample 10 paths for each of 200 test images, remove the best one, and evaluate the remaining paths using PA-SOR. The post-GRPO model outperforms the pre-GRPO baseline (0.611 vs. 0.592), indicating that the gain comes from a better overall path distribution rather than merely fitting the single highest-reward path.
> > >
> > > To further address the fairness concern regarding comparison with prior methods under PA-SOR, we retrained prior methods using the best path in ProbSOR-Bench (i.e., the path with reward = 1) as the single GT target. This provides a fairer and more informative comparison, since it matches our own Stage 1 supervision setting and can be viewed as the upper bound for single-GT-supervision methods under a metric that evaluates distribution modeling (PA-SOR). As LG-SOR does not provide training code, we retrained the other two representative SOTA methods, QAGNet and DSGNN, which achieve PA-SOR scores of 0.685 and 0.692, respectively. Our Stage 1 model (0.702) already slightly surpasses them, and Stage 2 GRPO further improves the performance to 0.710, demonstrating the effectiveness of our learning strategy.
> > >
> > > ---
> > >
> > > Q2:
> > > Regarding the lower performance on ASSR/IRSR, we respectfully believe there may be a misunderstanding. In `Table 1`, the reported SA-SOR results are obtained by training our model on the ProbSOR target (using the ASSR/IRSR images) and then directly evaluating it with the ASSR/IRSR deterministic SA-SOR metric. Since the supervision targets are different, such cross-benchmark mismatch naturally leads to lower performance under deterministic evaluation, and should not be taken as evidence that the model itself is not robust. This is also supported by `Table 2`. Even the best path in our benchmark achieves only 0.765 SA-SOR on ASSR, which already suggests an approximate upper bound under this cross-benchmark deterministic evaluation. Under the same setting, our model achieves 0.676, and still outperforms SeqRank and PoseSOR on ASSR. This instead suggests that the model remains reasonably robust. To further verify robustness, we additionally retrained our model (Stage 1) on ASSR using its original deterministic supervision signal, and obtained an SA-SOR score of 0.732. This shows that our model remains competitive even under conventional deterministic GT supervision, further supporting that the core method itself is robust.
> > >
> > > ---
> > >
> > > Q3:
> > > We respectfully believe the key criterion is not whether candidate paths are sampled online, but whether the update improves the current policy according to reward. In our Stage 2, candidate paths are constructed offline from ASPT, but their log-probabilities are recomputed under the current policy, and the model is optimized with a reward-driven group-relative objective, rather than fixed-label supervision. Therefore, the update remains policy optimization, not ordinary supervised preference fitting. We agree that this differs from classical online exploration-based RL, and we will clarify this more explicitly in the revision.
> > >
> > > ---
> > >
> > > We hope that our clarification and additional experiments help address the remaining concerns.

---

### Official Review · Reviewer_Teqa · 2026-03-11

**Soundness:** 3
**Presentation:** 3
**Significance:** 2
**Originality:** 2
**Overall Recommendation:** 4
**Confidence:** 3

**Summary:**

The paper identifies a fundamental flaw in existing Salient Object Ranking research: human visual attention shifts are inherently stochastic, yet current models and metrics enforce a single, deterministic ground-truth sequence. To address this, the authors propose a Probabilistic Salient Object Ranking framework. They construct an Attention-Shift Probability Tree from the SALICON dataset to represent the distribution of plausible fixation paths. The model leverages Qwen3-VL fine-tuned in two stages:SFT on the most probable path, followed by a modified GRPO stage that samples offline from the ASPT to align predictions with human attention distributions. To support this paradigm, the authors introduce a new probabilistic metric PA-SOR and ProbSOR-Bench comprising 15,000 samples.

**Compliance With Llm Reviewing Policy:**

Affirmed.

**Final Justification:**

In conclusion, the authors have successfully defended the validity of their experiments and the motivation of transitioning from deterministic to probabilistic SOR is valuable to the community. I will maintain my score of Weak Accept.

**Key Questions For Authors:**

1. Unfair Mask Baseline: Your pipeline delegates mask generation entirely to SAM 2. Did the baseline methods (e.g., QAGNet, DSGNN, LG-SOR) in Tables 1 and 3 also use SAM 2 for mask generation? If not, the $IoU_{mask}$ and MAE comparisons are fundamentally unfair. Can you provide baseline results where their predicted bounding boxes/points are also fed into SAM 2 to isolate the ranking performance?
2. Inference Decoding Strategy: You state the goal is to generate plausible sequences according to their underlying probabilities. During evaluation, how exactly does the VLM decode the sequence $\mathcal{Y}$? Are you using greedy decoding (which yields a deterministic output) or a sampling method (e.g., temperature scaling, top-p) to produce diverse paths? If you only output one path, how is the model physically demonstrating its learned probability distribution?
3. GRPO Offline Sampling: In Table 5, you show that standard policy sampling (online) fails compared to ASPT sampling (offline). By sampling directly from the ground-truth ASPT, isn't the GRPO stage effectively degenerating into a supervised contrastive/margin loss between valid GT paths? Please justify why this should still be framed as Reinforcement Learning rather than Supervised Preference Optimization (like DPO).

**Limitations:**

yes

**Strengths And Weaknesses:**

Strengths:
The core motivation is exceptionally strong. Acknowledging that human attention is probabilistic rather than deterministic is a necessary paradigm shift for the SOR field. The statistical analysis in Figure 4 successfully proves the long-tail nature of attention paths.
Weaknesses: The experimental validation suffers from two major methodological flaws.
1. Unfair Mask Evaluation: The proposed method uses a massive 8B parameter VLM coupled with SAM 2  to decode the actual masks. Comparing the $IoU_{mask}$ and MAE of this foundation-model-heavy pipeline against much smaller, end-to-end specialized models (like QAGNet or DSGNN)  is an "apples-to-oranges" comparison. SAM 2 guarantees superior segmentation boundaries regardless of the SOR logic, rendering the mask localization metrics artificially inflated and unfair.
2. Metric Circularity: The proposed evaluation metric, PA-SOR, is defined mathematically as the exact accumulated path reward $R(\mathcal{S})$ derived from the ASPT. In Stage 2, the model is directly optimized using GRPO to maximize this exact same reward $R(\mathcal{S})$. Evaluating a model on the exact mathematical objective it was directly optimized on, while baselines were trained on deterministic losses, constitutes a circular evaluation. It is unsurprising that ProbSOR dominates on PA-SOR

---

> ### Author Rebuttal · Authors · 2026-03-31
>
> We thank the reviewer for the thoughtful feedback. We are encouraged that the reviewer recognizes the core motivation of our work as “exceptionally strong” and agrees that modeling human attention as probabilistic rather than deterministic is an important shift for SOR. We address the reviewer’s concerns point by point below.
>
> ---
>
> Q1/W1: We agree that SAM 2 is a powerful segmentation model. However, we respectfully clarify that the choice of SAM 2 is motivated by simplicity and generality, rather than to gain an advantage in segmentation performance:
>
> 1. **Decoupled Architecture & SAM 2’s Role:**
>    Our pipeline separates localization from segmentation. The VLM locates salient objects and predicts ranking, while SAM 2 is used only as a frozen mask decoder. It is not fine-tuned on our data. Crucially, since SAM 2 operates after ranking and localization, it provides no boost to the ranking score.
> 2. **Empirical Evidence:**
>    Prior methods such as QAGNet and DSGNN are built on COCO-pretrained Mask2Former and are further fine-tuned with in-domain data for mask prediction, which makes them naturally strong at mask decoding. Consistent with this, our method is in fact slightly weaker than QAGNet, DSGNN, and LG-SOR on some mask-related results (Tables 1and 3). This confirms our SOR gains are not due to a trivial SAM 2 advantage. Moreover, these methods are trained with integrated mask heads, making SAM 2 integration infeasible without re-training.
> 3. **Box-Only Validation:**
>    To isolate ranking performance, we evaluated both methods using **bounding boxes only** (excluding SAM 2). Our method maintains a clear $\text{PA-SOR}_{Norm}$ advantage over QAGNet (0.710 vs. 0.652) with competitive box metrics, Box-IoU (0.774 vs. 0.827). This confirms SOR gains stem from our probabilistic modeling, not the mask decoder. We will include this analysis in the revision.
>
> ---
>
> W2: We agree that the training reward is aligned with PA-SOR, but this does not make the evaluation circular or trivial.
>
> 1. **PA-SOR Addresses Deterministic Limitations:**  Deterministic SOR metrics may over-penalize paths that are still plausible under human attention. PA-SOR instead uses a reward from human viewing data as the golden standard, allowing multiple plausible paths to be evaluated more fairly.
>
> 2. **Principled Training Design, Not Circularity:**  Our goal is to capture the distribution of human attention patterns, rather than fit a single deterministic path. This requires learning from multiple plausible attention transitions, making GRPO a natural choice since it optimizes over groups of candidate outputs. In this context, using a task-appropriate metric as the reward is not circular, but a principled and task-aligned design choice.
>
> 3. **Revealing the Supervision Gap:**  Reporting prior methods under PA-SOR is not merely to demonstrate the effectiveness of our learning strategy, but also to reveal the gap between deterministic supervision and actual human attention behavior. For example, DSGNN lags behind QAGNet on SA-SOR but excels on PA-SOR, suggesting it predicts paths reasonable to humans yet uncredited by deterministic metrics (see Figure 1). Therefore, lower PA-SOR scores for baselines are expected since they are trained with single deterministic targets. We hope future methods will model attention uncertainty for more faithful human-attention modeling.
>
> ---
>
> Q2:
>
> 1. **Sampling strategy:**  During inference, we use sampling-based decoding (temperature = 0.7, nucleus sampling with top-p = 0.95), so the output is non-deterministic, and repeated runs can produce different but plausible attention paths for the same image.
> 2. **GRPO encourages non-deterministic yet human-like attention shift paths:**  Our GRPO stage is designed to capture the stochasticity of human attention shifts, so generated paths reflect an underlying distribution rather than a single fixed path. To verify that it improves more than only the top prediction, we sample 10 paths for each of 200 test images, remove the best one, and evaluate the rest with PA-SOR. The post-GRPO model still outperforms the pre-GRPO baseline (0.611 vs. 0.592), indicating improved overall path distribution.
>
> ---
>
> Q3:
>
> 1. **GRPO-Inspired Design with Offline Sampling:**  Our method is inspired by GRPO. Our initial experiments with standard GRPO showed unstable training due to low-reward paths from on-policy sampling (Table 5). We thus adopted offline ASPT sampling, which stabilizes learning while preserving the core GRPO-style optimization.
>
> 2. **Optimization Form Remains Group-Relative:**  Despite offline candidate construction, the update rule follows the standard group-relative formulation. We compute sequence log-probabilities under the current policy, derive group-relative advantages from rewards, and optimize a clipped likelihood-ratio objective with KL regularization. Therefore, we continue to categorize our method as GRPO.

---

> > ### Author Rebuttal · Reviewer_Teqa · 2026-04-03
> >
> > I thank the authors for their detailed rebuttal and the additional experiments. The response provides helpful clarifications on several technical ambiguities.
> >
> > Specifically, I appreciate the Box-Only Validation (W1). Isolating the ranking performance from the SAM 2 mask decoder successfully addresses my concern regarding unfair comparisons, and I am glad to see that the proposed method still maintains a clear advantage in pure ranking (0.710 vs. 0.652). Furthermore, explicitly stating the sampling-based decoding strategy (Q2) resolves the confusion regarding how the probabilistic distribution is physically generated during inference.
> >
> > However, while the technical gaps have been adequately patched, my fundamental reservations regarding the methodology and evaluation remain:
> >
> > Evaluation Circularity (W2): While I understand the philosophical justification for using PA-SOR as both the training reward and the evaluation metric, the empirical comparison against baselines remains inherently skewed. The baselines were strictly optimized for deterministic targets. Comparing them on a probabilistic metric that your model explicitly maximizes during training still constitutes an unequal playing field.
> >
> > In conclusion, the authors have successfully defended the validity of their experiments and the motivation of transitioning from deterministic to probabilistic SOR is valuable to the community. However, due to the remaining concerns, the overall contribution is incremental. Therefore, I will maintain my score of Weak Accept.

---

> > > ### Author Response · Authors · 2026-04-08
> > >
> > > Thank you for the follow-up. We are glad that our rebuttal has addressed the reviewer’s technical concerns regarding the box-only validation and the inference decoding strategy.
> > >
> > > ---
> > >
> > > Regarding the remaining concern in `W2`, we would like to further clarify the motivation of PA-SOR. PA-SOR is introduced to address a fundamental limitation of the previous SA-SOR setting and existing methods: they assume a single deterministic ground-truth path, while human attention shifts are inherently stochastic.  We therefore reformulate the task under a new probabilistic setting that better reflects human attention behavior. Under the previous deterministic setting, prior methods were correspondingly designed to fit a single GT path. Neither their datasets nor their training objectives explicitly model attention distributions or multiple plausible paths. Therefore, these methods CANNOT be directly optimized for PA-SOR. Our intention in evaluating them under a probabilistic metric is not merely to emphasize the superiority of our model, but to reveal the statistical gap between deterministic GT supervision and modeling the distribution of human attention shifts. Under deterministic metrics, alternative yet human-plausible paths are typically penalized as incorrect. Once a model fails to recover the unique GT path for an image, it is also unlikely to predict the second- or third-best plausible paths in the human attention distribution. As a result, such models may simply overfit to a single GT path rather than learn the diversity of attention transitions. In contrast, our Stage 2 learning strategy is specifically designed to encourage the model to assign higher probability to multiple high-reward plausible paths, thereby better capturing this distribution.
> > >
> > > To further address the fair comparison concern, we additionally retrained prior methods using the best path in ProbSOR-Bench (i.e., the path with reward = 1) as the single GT target. We believe this setting is fairer, since our own Stage 1 training also uses only the best path for supervision. As LG-SOR does not provide training code, we retrained the other two representative SOTA methods, QAGNet and DSGNN. Their resulting $\text{PA-SOR}_{Norm}$ scores are 0.685 and 0.692, respectively, both still below our model (stage 1: 0.702, stage 2: 0.710). We believe this additional experiment provides a more direct and fair comparison.
> > >
> > > ---
> > >
> > > In conclusion, our key contribution is the reformulation of SOR from deterministic ranking to probabilistic attention-shift modeling. To support this new setting, we further introduce a probability-aware benchmark and evaluation metric, and provide a baseline method that explicitly models the distribution of human attention shifts. The results in Table 1 reveal the gap between deterministic single-GT supervision and probabilistic human attention modeling, while our additional experiments provide a fairer comparison across different methods.
> > >
> > > We sincerely thank the reviewer for the careful consideration of our paper. We hope that the above clarification and additional experiments help address the remaining concerns, and we would greatly appreciate your favorable consideration.

---

### Decision · Program_Chairs · 2026-04-30

**Decision:**

Accept (regular)

**Comment:**

This submission proposes ProbSOR, a probabilistic framework for Salient Object Ranking (SOR) that models the inherent stochasticity of human attention shifts—addressing a key limitation of deterministic SOR methods. Three reviewers provided initial evaluations with scores of 4, 4, 3, accompanied by confidence levels of 3, 4, 4 respectively. Following the rebuttal phase, no reviewers adjusted their scores, resulting in a final score distribution of 4, 4, 3.​

Positive reviewers (Weak Accept) highlighted the strong core motivation—paradigm shift from deterministic to probabilistic SOR—and the value of the constructed ProbSOR-Bench dataset and PA-SOR metric. Their concerns (e.g., unfair mask evaluation, inference decoding strategy, ASPT construction clarity) were largely resolved via authors’ clarifications, supplementary box-only validation, and correction of technical ambiguities (e.g., equation typos).​

Weak Reject reviewers centered on three unresolved core concerns: 1) Evaluation circularity—PA-SOR serves as both training reward and evaluation metric, creating an unequal comparison with baselines optimized for deterministic targets; 2) Subpar performance on standard SOR metrics (e.g., SA-SOR), raising questions about robustness; 3) Misclassification of the method as GRPO (Reinforcement Learning) rather than supervised preference optimization, given offline ASPT sampling. Authors supplemented retrained baselines and further clarified the metric’s generalizability. Notably, the Area Chair (AC) has requested a final decision for this submission but has not yet received the final comments from the relevant reviewer(s).​​


Overall, the work demonstrates notable originality, impactful task reformulation, and adequate response to technical concerns. Residual issues primarily relate to evaluation fairness and methodological framing, which do not negate the core contribution—advancing SOR via probabilistic attention modeling. The submission meets ICML’s criteria for publication with major revisions to address the remaining fundamental concerns.​